# Associations between Postprandial Gut Hormones and Markers of Bone Remodeling

**DOI:** 10.3390/nu13093197

**Published:** 2021-09-14

**Authors:** Nina Wittorff Jensen, Kim Katrine Bjerring Clemmensen, Marie Møller Jensen, Hanne Pedersen, Kristine Færch, Lars Jorge Diaz, Jonas Salling Quist, Joachim Størling

**Affiliations:** 1Clinical Prevention Research, Steno Diabetes Center Copenhagen, 2820 Gentofte, Denmark; kim.katrine.bjerring.clemmensen.01@regionh.dk (K.K.B.C.); marie.moeller.jensen@regionh.dk (M.M.J.); hanne.pedersen.12@regionh.dk (H.P.); kristine.faerch@regionh.dk (K.F.); jonas.salling.quist@regionh.dk (J.S.Q.); 2Department of Clinical Medicine, Aalborg University, 9000 Aalborg, Denmark; 3Department of Biomedical Sciences, University of Copenhagen, 1165 Copenhagen, Denmark; joachim.stoerling@regionh.dk; 4Clinical Epidemiology Research, Steno Diabetes Center Copenhagen, 2820 Gentofte, Denmark; lars.jorge.diaz@regionh.dk; 5Translational Type 1 Diabetes Research, Clinical Research, Steno Diabetes Center Copenhagen, 2820 Gentofte, Denmark

**Keywords:** bone markers, gut hormones, bone metabolism, CTX, P1NP

## Abstract

Gut-derived hormones have been suggested to play a role in bone homeostasis following food intake, although the associations are highly complex and not fully understood. In a randomized, two-day cross-over study on 14 healthy individuals, we performed postprandial time-course studies to examine the associations of the bone remodeling markers carboxyl-terminal collagen type I crosslinks (CTX) and procollagen type 1 N-terminal propeptide (P1NP) with the gut hormones glucose-dependent insulinotropic polypeptide (GIP), glucagon-like peptide 1 (GLP-1), and peptide YY (PYY) using two different meal types—a standardized mixed meal (498 kcal) or a granola bar (260 kcal). Plasma concentrations of total GIP, total GLP-1, total PYY, CTX, and P1NP were measured up to 240 min after meal intake, and the incremental area under the curve (iAUC) for each marker was calculated. The iAUC of CTX and P1NP were used to assess associations with the iAUC of GIP, GLP-1, and PYY in linear mixed effect models adjusted for meal type. CTX was positively associated with GIP and GLP-1, and it was inversely associated with PYY (all *p* < 0.001). No associations of P1NP with GIP or GLP-1 and PYY were found. In conclusion, the postprandial responses of the gut hormones GIP, GLP-1, and PYY are associated with the bone resorption marker CTX, supporting a link between gut hormones and bone homeostasis following food intake.

## 1. Introduction

Bone remodeling is a highly dynamic process that takes place throughout life and helps maintain the skeleton [1]. Circulating bone remodeling biomarkers can be used to assess bone metabolic status [2]. Carboxyl-terminal collagen type I crosslinks (CTX) is an indicator of bone resorption, where osteoclasts breaks down the bone tissue, whereas procollagen type 1 N-terminal propeptide (P1NP) is an indicator of bone formation, where osteoblasts build up the bone tissue [3,4]. Studies have shown a circadian pattern for bone remodeling, with high bone resorption taking place during nighttime and high bone formation taking place during daytime, correlating with the highest CTX concentration in the morning after an overnight fast [5,6]. The bone remodeling marker levels also change acutely following food intake, where CTX is reduced and P1NP is increased, thus reflecting increased bone formation [7,8].

The gut hormones glucose-dependent insulinotropic polypeptide (GIP) and glucagon-like peptide-1 and 2 (GLP-1/2) are secreted postprandially by the intestinal K and L cells, respectively [9]. Accumulating evidence suggests that these hormones and especially GIP are involved in bone remodeling [10,11]. Recent studies have shown that the intravenous infusion of GIP acutely decreases CTX plasma levels, which is suggestive of increased bone formation [12,13]. Treatment with the GLP-1 receptor agonist liraglutide has been shown to increase P1NP levels and to maintain bone mineral density in obese women after weight loss, suggesting a positive effect of GLP-1 on bone formation [14]. Interestingly, infusion with GLP-1 acutely suppresses circulating CTX levels similar to GIP, but when co-infused, GLP-1 plus GIP seem to have partially synergistic effects on CTX [14]. The appetite-inhibiting hormone Peptide YY (PYY) is co-secreted postprandially from the intestinal L-cells with GLP-1/2 and is believed to mainly have bone resorptive effects [6]. Thus, gut hormones seem to represent an important link between food intake and bone homeostasis; however, the complexity of the associations between different gut hormones and bone turnover markers following food intake remains elusive.

In this study, we aimed to further investigate the associations between the bone remodeling markers CTX and P1NP and the gut hormones GIP, GLP-1, and PYY following the intake of two types of meals in healthy individuals. 

## 2. Materials and Methods

### 2.1. Study Design

This study is based on the randomized, open-label, cross-over study, PRESET. The primary objective in PRESET was to assess gastrointestinal (GI) transit time in response to either a standardized mixed meal or a granola bar (SmartBar^®^) using a wireless capsule technique (SmartPill^®^) (data are presently unpublished). The PRESET study was conducted before the ongoing RESET study [15], in which the SmartPill^®^ is used in combination with a standardized mixed meal. A power calculation has been made for the PRESET study, with the main focus being on investigating gastrointestinal transit patterns with a wireless motility capsule. The number of participants needed for this study was calculated to be 12.3, and to cope with possible dropouts, 15 participants were recruited, 14 of which completed the study. All of participants were recruited through an advertisement in a free newspaper in the Copenhagen area and trough an online Danish recruitment website (https://www.forsoegsperson.dk, accessed on 1 September 2018). The inclusion criteria were to be between 30 and 70 years of age and to have a body mass index (BMI) from 18.5 to 25 kg/m^2^. The exclusion criteria were a self-reported history of an eating disorder in the past three years, a weight change above 5 kg within the last three months, HbA1C above 39 mmol/mol, uncontrolled medical issues, diabetes or other endocrine diseases, immunosuppression, current treatment with medication or medical devices that affectGItransit time, current treatment with medications that affect glucose metabolism or appetite, peroral steroids, non-steroidal anti-inflammatory drugs, tricyclic antidepressants, selective serotonin re-uptake inhibitors or opioids, bariatric surgery, GI surgery within 3 months prior to inclusion, GI diseases or regular (weekly) GI symptoms such as abdominal pain, dysphagia, diarrhea, etc., alcohol/drug abuse or in treatment with disulfiram (Antabus) at time of inclusion, pregnancy or lactation, or concomitant participation in other research studies. The study was conducted at the Steno Diabetes Center Copenhagen, Gentofte, Denmark, from January to May 2019. The participants were 14 healthy adults with a BMI between 18.5 and <25 kg/m^2^. They were instructed to fast the evening before from 8 pm prior to each of the two study days. The participants were instructed not to engage in any strenuous physical activity or to consume alcohol 48 h prior to the study days. Based on postprandial time-course studies by others [2,16], we chose to collect blood samples through a venous catheter before and at 15, 30, 45, 60, 90, 120, 180 and 240 min after consuming a meal on the two study days. Such a detailed and long-term postprandial time-course would allow us to obtain a detailed view of the changes in the gut hormones and bone markers. The only difference between the two study days was the type of meal, which was either a granola bar (260 kcal, 7E% fat, 74E% carbohydrate, and 19E% protein) (SmartBar^®^, Medtronic, MN, USA) or a standardized mixed meal consisting of yoghurt, muesli, rye bread, wheat bread, cheese, and butter and jam (498 kcal, 34E% fat, 49E% carbohydrate, and 17E% protein). Both meals were consumed with 200 mL of water within 10 min. The order of the meals was randomized, and the mean period between the study days was 24 days.

The study was performed in accordance with the ethical standards of the institutional and national research committee (H-18026293) and with the 1964 Declaration of Helsinki and its later amendments. The study is registered at ClinicalTrials.gov (ClinicalTrials.gov identifier: NCT03894670). All participants gave written informed consent prior to any testing, and they all gave renewed consent for the analysis of the bone markers. 

### 2.2. Measurements

All plasma samples were collected, handled, and analyzed at the Steno Diabetes Center Copenhagen, Gentofte, Denmark. All samples were analyzed in duplicate, and the mean and the coefficient of variation (CV%) were calculated. Blood was collected in ethylenediaminetetraacetic acid (EDTA) sample-tubes with the dipeptidyl peptidase IV (DPP4) inhibitor valine-pyrrolidide (1 mM) added, and the samples were kept on ice. The sample-tubes were centrifuged at 3500× *g* for 15 min at 4 °C. Plasma was stored at −80 °C until further analysis. The bone markers CTX and P1NP were measured on a Cobas 6000 e601 (Roche Diagnostic, IN, USA) using an electrochemiluminescence immunoassay. Intra- and interassay variations for CTX and P1NP were calculated on control material from Roche at two levels—a low- and a high-concentration control sample [17]. The intra- and interassay CV% for both analytes in the control samples were 0.9–3.0%. The gut hormones, GIP, GLP-1, and PYY, were also measured by an electrochemiluminescence immunoassay method using multiplex technology (Meso Scale Discovery [MSD], Rockville, MD 20850, USA) on a MESO QuickPlex SQ 120 instrument (Model.No.: 1300). The samples were measured according to the manufacturer’s recommendations. Before analyses, the samples were thawed on ice and were centrifuged at 2000× *g* for 3 min at 4 °C. The total GIP, total GLP-1, and total PYY were analyzed in duplicate by the high sensitivity U-PLEX Metabolic Group 1 (Human) Multiplex Assay Kit (Kit LOT# 309670 and 309671). DPP-IV Inhibitor (MSD LOT# SLBX4085) was added to prevent degradation of GIP, GLP-1, and PYY, as these are cleaved by DPP-IV [18,19]. Intra- and interassay variations for GIP, GLP-1, and PYY were estimated on internal plasma control pools. The intra-/inter-assay variations were as follows: GIP: 17.3%/51.8%, GLP-1: 3.2%/13.6% and PYY: 4.1%/17.9%.

For 77–100% of the samples measured, the CV% between the duplicate values was under or equal to 20% for all five biomarkers. All duplicate sample results for CTX and P1NP and 99.2% of all of the PYY duplicate values had a CV% under 20%. No sample values were below the detection limit.

### 2.3. Statistical Analyses

All of the available data were included in the analyses. Differences in the plasma levels of GIP, GLP-1, PYY, CTX, and P1NP between the two meal types were analysed at individual time points by paired *t*-test using the Benjamini–Hochberg correction. The incremental area under the curve (iAUC_0–240_) for GIP, GLP-1, PYY, CTX, and P1NP was calculated by first calculating the total area under the curve using the trapezoidal rule and by then subtracting the fasting values. The incremental area under the curve was used as a summary measure of the change in the hormones/markers following the meal tests. 

Linear mixed effect models with a participant-specific random intercept were used to investigate the associations of the iAUC’s of the two outcomes, CTX and P1NP, with the iAUC’s of the three gut hormones GIP, GLP-1, and PYY. GIP, GLP-1, PYY, and meal type were included as fixed effects in the two models. Visual inspection was used to assess the normality of the model residuals. The models were not adjusted further due to the limited number of participants. The outcome variables were, if needed, logarithmically transformed before analysis to obtain normally distributed model residuals. Data analysis was performed in R version 3.6.0. A two-sided 5% level of significance was used.

## 3. Results

The characteristics of the study participants are listed in Table 1, and Table 2 shows the fasting concentrations of glucose, insulin, the bone markers, and the gut hormones measured during the two study days prior to consumption of either a standardized mixed meal or a granola bar. 

After the consumption of either of the two meal types, the plasma concentrations of GIP, GLP-1, and PYY increased as expected (Figure 1A–C). The standardized mixed meal was associated with a slightly more pronounced postprandial response of all three gut hormones compared to the granola bar, as demonstrated by the iAUC analyses, which showed significantly higher postprandial levels of GIP and PYY for the mixed meal versus the granola bar (Figure 1A,C and Table 2). In response to both meal types, CTX, also as expected, decreased (Figure 1D). As for the gut hormones, the standardized mixed meal was associated with a more pronounced and prolonged decrease in CTX compared to the granola bar (Figure 1D and Table 2). There were no postprandial changes in the levels of P1NP after the intake of either meal type (Figure 1E and Table 2).

In linear mixed effect models adjusted for meal type, we investigated associations between the iAUC’s of the two outcomes, CTX and P1NP, and the three gut hormones GIP, GLP-1, and PYY. We found that GIP and GLP-1 were positively associated with CTX_iAUC_ (*p* < 0.001), whereas PYY_iAUC_ was inversely associated with CTX_iAUC_ (*p* < 0.001) (Table 3). None of the gut hormones were found to be associated with P1NP_iAUC_. 

## 4. Discussion

In this study, we investigated whether the postprandial plasma levels of the gut hormones GIP, GLP-1, and PYY are associated with the levels of the bone remodeling markers CTX and P1NP following the intake of two different meal types—a standardized mixed meal or a granola bar. We found associations between CTX and all three gut hormones, whereas no associations were found for P1NP.

Our results from the mixed effect models showed that GIP and GLP-1 were positively associated with CTX, whereas PYY was inversely associated with CTX. These findings are in accordance with previous studies reporting a relationship of GLP-1 and especially GIP with bone turnover markers, including CTX [11,13,20]. Our study thus confirms gut hormones as likely critical bone turnover regulators, thereby representing a link between food intake and bone homeostasis. As opposed to other studies, which primarily used GIP and/or GLP-1 infusions [11,13,20], we used two different mixed meal types containing different amounts of calories: 260 kcal for granola bar vs. 498 kcal for standardized mixed meal. The combined association analysis of both mixed meal types is a strength of our study. Another strength in this study is the repeated measurements in the same individual. 

Postprandial GIP plasma levels correlate with bone remodeling, and recent studies have examined whether the intravenous administration of GIP or a GIP receptor antagonist affects the postprandial plasma levels of bone turnover markers. The studies reported that infusion with GIP caused a decrease in CTX and an increase in P1NP [12,13,20], whereas GIP receptor antagonism partly reversed the postprandial decrease in CTX [20]. Another study reported that GIP in combination with high plasma glucose acutely lowers CTX [21]. GIP may therefore be the single most important gut hormone modulator of bone turnover following food intake, but more studies are needed to fully clarify and discriminate between the individual effects of different gut hormones in regulating bone homeostasis.

If gut hormones constitute a link between food intake and bone homeostasis, gut hormones may directly affect bone turnover via binding to receptors on the surface of the osteoblasts and/or osteoclasts. In line with this notion, both bone and osteoblast- and osteoclast-derived cell lines express the GIP receptor (GIPR), supporting the notion that GIP could have direct bone remodeling effects [6,22,23,24]. It remains to be established if osteoblasts and osteoclasts express the GLP-1 receptor, as opposing data exists [23]. However, as GLP-1 stimulates bone marrow stromal cell differentiation into osteoblasts, GLP-1 may promote bone formation indirectly by increasing the number of osteoblasts from marrow stromal cells [25]. Hence, opposed to the effect of GIP, which may directly stimulate osteoblast activity and, hence, bone formation, GLP-1’s effect on osteoblasts is mainly dependent on the formation of new cells, which is unlikely to have an acute bone formation effect. In this way, GLP-1 may set the scene for more pronounced and direct effects exerted by other bone remodeling hormones such as GIP, although this is purely speculative. 

We did not observe any postprandial changes in the P1NP levels following the intake of either meal type and found no statistically significant associations between the iAUC for P1NP and the three gut hormones. While some studies reported decreased P1NP levels after food ingestion, which is indicative of decreased bone formation [26,27,28], others also found a lack of postprandial P1NP changes following oral glucose tolerance testing or GIP infusion [11,20]. In these studies, however, the meals consisted of more calories (up to 2150 kcal) compared to the number of calories in the present study (260/498 kcal). This may indicate that more calories or a higher food volume is needed to cause a decrease in P1NP [26,27,28].

We did not measure bone markers other than CTX and P1NP, such as osteocalcin, in the current study. However, while some animal studies suggest a link between gut hormones and osteocalcin, there seems to be a general lack of clinical evidence for associations between gut hormones and osteocalcin [6,29].

The inverse association between CTX and PYY suggests that PYY may play a role in postprandial bone turnover via catabolic effects on bone. The idea that PYY is a negative regulator of osteoblastic bone formation is supported by studies in PYY receptor knockout mice, which demonstrated accelerated and increased bone growth and mass [30,31]. Studies in humans also previously reported inverse relationships between bone formation and PYY, thereby further linking PYY to bone homeostasis [32,33,34,35].

The limitations of our study include the small sample size (*n* = 14), the sex distribution skewing towards female, and that both pre- and postmenopausal women are included. Although these limitations may hamper our study in terms of generalizability, we believe our identified associations between postprandial gut hormones and bone markers add to the current knowledge of the gut–bone axis. Larger and more equally distributed studies should be performed to further elucidate the associations observed in our study. We believed the use of two different meal types makes our study somewhat more representative and physiologically relevant compared to most other studies investigating associations between gut hormones and bone markers. Another strength of our study is that we included multiple timepoints up to 240 min after food intake, allowing us to perform iAUC association analyses, which contrasts with many other studies, which only examined a single or a few postprandial timepoints. Hence, the current study may better reflect the normal postprandial physiology of the gut–bone axis.

## 5. Conclusions

Our study supports an association between gut hormones and the bone marker CTX after food intake, thereby adding to the evidence that gut hormones play an important role in the postprandial signal for bone formation when energy and nutrient supply are abundant.

## Figures and Tables

**Figure 1 nutrients-13-03197-f001:**
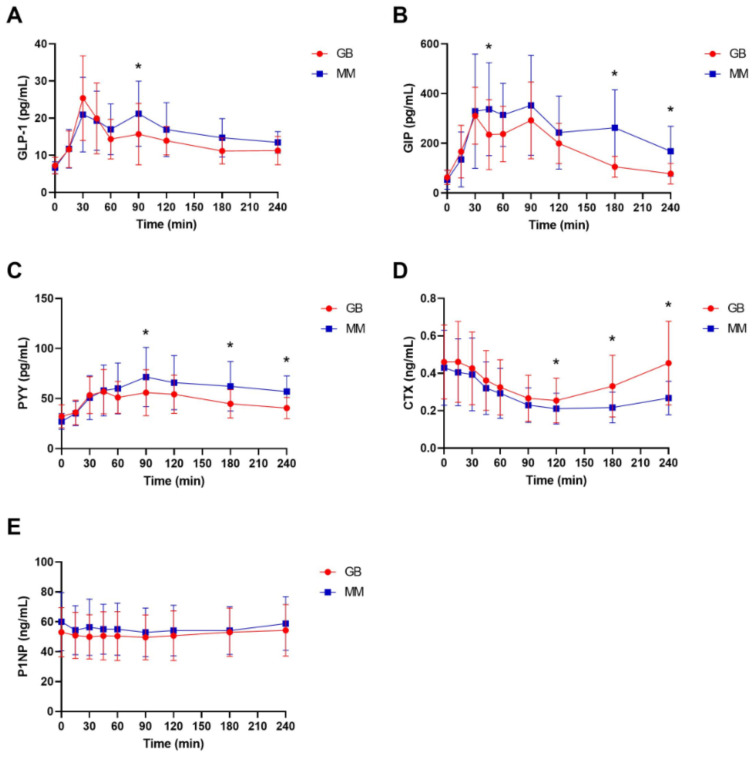
Mean (±SD, standard deviation) postprandial plasma concentrations of the gut hormones GLP-1 (glucagon-like peptide) (**A**), GIP (glucose-dependent insulinotropic polypeptide) (**B**), and PYY (peptide YY) (**C**) and the bone remodeling markers CTX (carboxyl-terminal collagen type I crosslinks) (**D**) and P1NP (**E**) for the two types of meals (granola bar and standardized mixed meal (MM)), *n* = 14. Significant differences between the two groups at individual time points are marked by * and denote *p* < 0.05. GB, granolabar, MM, mixed meal.

**Table 1 nutrients-13-03197-t001:** Characteristics of the study population.

	Overall
*n*	14
Age (years)	53.8 (45.8, 64.5)
Males *n* (%)	3 (21.4)
BMI (kg/m^2^)	23.1 (21.8, 23.9)
Weight (kg)	63.9 (59.9, 69.7)
HbA1c (mmol/mol)	34.5 (32.5, 36.0)

Data are medians (Q1; Q3) or *n* (%). Abbreviations: BMI, body mass index; HbA1c, Hemoglobin A1c, Q1, quartile 1; Q3, quartile 3.

**Table 2 nutrients-13-03197-t002:** Plasma concentrations of glucose, insulin, gut hormones, and bone markers in the study population during the two study days for each meal type.

	Standardized Mixed Meal	Granola Bar
*n*	14	14
Fasting glucose (mmol/L)	5.2 (0.4)	5.2 (0.4)
Fasting insulin (pmol/L)	27.0 (20.5, 33.0)	29.0 (23.2, 34.8)
Fasting P1NP (ng/mL)	60.8 (44.0, 78.6)	50.4 (43.4, 62.5)
Fasting CTX (ng/mL)	0.4 (0.3, 0.5)	0.4 (0.3, 0.6)
Fasting GLP-1 (pmol/L)	6.4 (5.6, 8.1)	7.5 (5.6, 8.5)
Fasting GIP (pg/mL)	34.0 (26.9, 63.4)	54.3 (44.1, 75.8)
Fasting PYY (pg/mL)	25.7 (20.7, 33.6)	29.8 (24.5, 39.2)
P1NP_iAUC_ (ng/mL × min)	−947.4 (−1911.1, −221.7)	−270.5 (−561.1, −176.1)
CTX_iAUC_ (ng/mL × min)	−34.9 (−44.9, −21.2)	−30.7 (−37.0, −17.6)
GLP-1_iAUC_ (ng/mL × min)	7.0 (4.7, 9.6)	5.0 (2.8, 6.9)
GIP_iAUC_ (ng/mL × min) *	41.2 (34.2, 56.9)	25.7 (18.8, 31.1)
PYY_iAUC_ (ng/mL × min) *	7.0 (3.7, 10.4)	2.4 (1.8, 6.0)

Data are presented as medians (Q1; Q3) or mean (SD, standard deviation). Abbreviations: P1NP, procollagen type 1 *n*-terminal propeptide; CTX, carboxyl-terminal collagen type I crosslinks; GLP-1, glucagon-like peptide 1; GIP, glucose-dependent insulinotropic polypeptide; PYY, Peptide YY; iAUC, incremental area under the curve. * denotes significant difference (*p* < 0.05) between meal types.

**Table 3 nutrients-13-03197-t003:** Estimates (95% CI) of the associations between the two outcomes (CTX_iAUC_ and P1NP_iAUC_) and GLP-1_iAUC_, GIP_iAUC_, and PYY_iAUC_.

	CTX_iAUC_		P1NP_iAUC_	
	Estimate (95% CI)	*p*	Estimate (95% CI)	*p*
**GLP-1_iAUC_**	6.97 (3.39; 10.54)	<0.001	17.59 (−298.93; 334.12)	0.913
**GIP_iAUC_**	0.51 (0.24; 0.77)	<0.001	0.69 (−23.48; 24.87)	0.955
**PYY_iAUC_**	−8.34 (−11.20; −5.59)	<0.001	−59.82 (−311.40; 191.76)	0.641

The models were adjusted for meal type (granola bar vs. standardized mixed meal) with the participants (*n* = 14) included as a random effect. Units are ng/mL × min^−1^. Abbreviations: CTX_iAUC_, incremental area under the curve for CTX, P1NP_iAUC_, incremental area under the curve for P1NP, GLP-1_iAUC_, incremental area under the curve for GLP-1, GIP_iAUC_, incremental area under the curve for GIP, PYY_iAUC_, incremental area under the curve for PYY.

## Data Availability

The data that support the findings of this study are available from the corresponding author upon reasonable request.

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
