# Peer review of "Associations between Postprandial Gut Hormones and Markers of Bone Remodeling"

_nutrients, 2021, doi:10.3390/nu13093197_

Round 1

Reviewer 1 Report

The authors show an association between postprandial gut hormones and markers of bone remodelling in subjects after consumption of two different meals. This have been shown in earlier studies but with fewer timepoints.

-Please include power calculations for the number of study participants

-Please indicate statistics used for the base line measurements in table 2

Reviewer 2 Report

The work by Jensen et al is well articulated and topic of interest. With the human subject's data, they made a compelling case for their hypothesis. However, there is major revision required at this present form.

Major concern:

  1. Sample size: Significantly smaller sample size, inclusion criteria for the subjects needed to be described. To make the study unbiased different age groups with different sexes are required.
  2. Diet: Different dietary constituents can significantly alter the secretion of a host of different hormones which essentially can exhibit anabolic and catabolic implications. Therefore, the two different diets mentioned in the study should be explained a) what are their composition 2) what are the calorific values of those compositions

If possible incorporate diets such as high carbohydrate, high protein, and high fat to the subjects and explore the change in the level of GIP, GLP1,CTX and PINP. This could significantly enrich the information in the manuscript.

  1. Statistical significance is not indicated in the figure.
  2. What is the postprandial effect of other bone formation markers such as osteocalcin, alkaline phosphatase? Whether they have any associations with gut hormones such as GIP, GLP1/2? Please incorporate those aspects in the manuscript. In addition to evaluation of serum level, a secondary method of validation will be beneficial to solidify the authors’ claim.
  3. Is there any previous evidence/reference to monitor the timepoints until 240 mins? If authors can incorporate the reasoning why they included that could render a better perspective to the readers.
  4. Reference needed:

Line 172-175

Restructuring of the writing needed: Lines 194-196
